# Chylopericardium Effusion in a Lac Alaotra Bamboo Lemur (*Hapalemur alaotrensis*)

**DOI:** 10.3390/ani11020536

**Published:** 2021-02-19

**Authors:** Mara Bagardi, Jessica Bassi, Angelica Stranieri, Vanessa Rabbogliatti, Daniela Gioeni, William Magnone, Claudio Pigoli

**Affiliations:** 1Department of Veterinary Medicine, University Veterinary Teaching Hospital, University of Milan, 26900 Lodi, Italy; jessica.bassi@unimi.it (J.B.); angelica.stranieri@unimi.it (A.S.); vanessa.rabbogliatti@unimi.it (V.R.); daniela.gioeni@unimi.it (D.G.); claudio.pigoli@unimi.it (C.P.); 2Parco Natura Viva, Garda Zoological Park, 37012 Bussolengo, Italy; william.magnone@parconaturaviva.it; 3Laboratorio di Istologia, Sede Territoriale di Milano, Istituto Zooprofilattico Sperimentale della Lombardia e dell’Emilia-Romagna (IZSLER), 20133 Milano, Italy

**Keywords:** chylopericardium, lymphoCT, zoo animals

## Abstract

**Simple Summary:**

The clinical staff of the Veterinary Teaching Hospital of the University of Milan in Lodi have worked closely with veterinarians operating in a famous zoological park in Northern Italy. Thanks to this collaboration it has been possible to properly manage the clinical case reported in this brief communication. In veterinary medicine, clinical cases of chylopericardium in the species *Hapalemur alaotrensis* have never been described in the literature. Even in more common species, such as dogs and cats, it is a very rare pathology. The description of this clinical case and its diagnostic management can be a valid clinical and anatomopathological support for other colleagues who face a similar case, so that they can focus on diagnostic investigations useful for diagnosis. A new anesthesiologic protocol, allowing the optimal management of the patient throughout the clinical procedure, is also reported in detail, as well as the detailed description of the lymphoCT examination. For all these reasons the authors think that the description of this clinical case can help other colleagues in the management of infested subjects.

**Abstract:**

An 11-year-old female *Hapalemur alaotrensis* was evaluated following a history of dyspnea of 15 days’ duration. Thoracic radiography performed by the referring veterinarian revealed a large cardiac silhouette and dorsal deviation of the trachea. Heart sounds were muffled. Echocardiographic findings were indicative of severe pericardial effusion without cardiac tamponade. No pleural effusion was identified. A computed tomography (CT) exam confirmed the presence of severe pericardial effusion and allowed identification of a parenchymatous mediastinal lesion sited at the level of the left hemithorax. To delineate the thoracic duct, lymphoCT was also performed by injection of iodinated contrast medium in the perianal subcutaneous tissue. Pericardiocentesis yielded a considerable amount of effusion with chylous biochemical and cytological properties. A diagnosis of chylopericardium with absence of pleural effusion was made. Initially, the chylopericardium was managed conservatively with two centesis and oral treatment with prednisolone. Medical treatment did not result in complete resolution of effusion and clinical signs; therefore, subtotal pericardiectomy and thoracic duct ligation were recommended. After the second pericardiocentesis, the subject died and the pericardiectomy could not be performed. To the authors’ knowledge, this is the first report of the development of chylopericardium in a *Hapalemur alaotrensis*.

## 1. Introduction

Chylopericardium generally occurs after thoracic surgery or trauma. Primary idiopathic chylopericardium is a rare condition, especially in captive zoo animals. A few case reports have been published on human primary idiopathic chylopericardium caused by leakage via an acquired fistula into the pericardium [1,2,3].

In human medicine, primary chylothorax and/or chylopericardium is rare and is mostly observed after mediastinal surgery. Although the exact pathophysiology of primary chylopericardium has not been established, the reflux of chylous fluid into the pericardial space was charged as a primary cause. Damage to the thoracic duct valves, and communication with the pericardial lymphatics or abnormally elevated pressure in the thoracic duct, could cause chylous fluid reflux [4,5]. Furthermore, the etiology of chylopericardium is like that of idiopathic chylothorax [6]. Normally, the mesenteric lymphatic vessels unite to form the cisterna chyli. From there, lymph is channeled cranially through the diaphragm, continuing cranially into the thoracic duct that drains into the cranial vena cava, and the lymph enters the systemic circulation [6]. In case of chylothorax, lymph leaks from the lymphatic vessels in the thorax and is free in the pleural space. In idiopathic chylothorax, this is due to some abnormalities, such as lymphangiectasia [6]. Presumably, the cause of chylopericardium is similar, with the exceptions that the abnormal lymphatic vessels are contained within the pericardium, and that there is a connection between the thoracic duct and the pericardial lymphatic system. In humans, it has been suggested that the etiology of primary chylopericardium involves lymphangiectasia, the presence of an anatomic communication between the thoracic duct and the lymphatic vessels of the pericardial sac, and damage to the valves in the thoracic duct [5,7,8,9]. It is possible that chylopericardium is a self-perpetuating condition because pericardial effusion may lead to an increase in right-sided venous pressure, which would prevent the forward flow of chylous into the cranial vena cava. 

The diagnostic algorithm of chylopericardium includes, primarily, a chest x-ray that shows an enlarged cardiac silhouette, and a subsequent echocardiography, which can explain the cause of cardiomegaly and pericardial fluid. Chest radiography and echocardiography are the first-line diagnostic tools with the addition of CT—useful to rule out any mediastinal disease that could cause compression and obstruction of the thoracic duct. Furthermore, imaging modalities that have been used to evaluate humans with primary idiopathic chylopericardium also include lymphangiography and lymphoscintigraphy [1,10]. However, the final diagnosis could be performed only by pericardial fluid analysis. 

Some patients do not need surgical treatment, and they can be treated with conservative therapy and effective drainage, but a failure rate of 57–60% has been reported. In fact, the data presented by Reeves et al. (2019) provide no strong evidence in the literature for the use of medical therapy as a primary treatment method for idiopathic chylopericardium in dogs and cats [11]. Thoracic duct ligation and a pericardial window are the most effective procedures to prevent recurrence [7,12]. According to Akamatsu et al. (1994) [7], ligation and resection of the thoracic duct and construction of a pericardial window is the most effective treatment to prevent the recurrence of chylous fluid accumulation. In addition, it is difficult to draw strong conclusions regarding the effectiveness of any one surgical treatment for idiopathic chylopericardium rather than another in either dogs or cats. However, the best available evidence at this time is in dogs, and, although the evidence is limited, it appears to support surgery with either thoracic duct ligation and cisterna chyli ablation or thoracic duct ligation and subtotal pericardiectomy [11].

## 2. Materials and Methods

An 11-year-old female *Hapalemur alaotrensis*, weighing 1.7 kg, born and raised in captivity at “Natura Viva zoological park” in Bussolengo (VR, Italy), was presented at the Veterinary Teaching Hospital of the University of Milan for dyspnea, tachypnea, and open-mouth breathing. No history of trauma or infection was reported by the keeper. The zoo veterinarian reported a chronic disease course of three weeks, which initially presented with lethargy, decreased activity, weakness, and malaise. Physical examination showed respiratory distress, hypotension, and muffled heart sounds. To perform clinical procedures, general anesthesia was necessary. For this reason, it was not possible to record the basal vital parameters of the subject. 

Intramuscular premedication, with dexmedetomidine at 20 μg kg^−1^ (Dexdomitor^®^ 0.5 mg mL^−1^, Vetoquinol, Bertinoro (FC), Italy), ketamine at 5 mg kg^−1^ (Lobotor, 100 mg mL^−1^, Acme srl, Cavriago (RE), Italy), and methadone at 0.2 mg kg^−1^ (Semfortan^®^ 10 mg mL^−1^, Dechra Veterinary Products, Torino, Italy), was performed. After premedication, heart rate was 80 bpm, breathing rate varied between 40 and 50 bpm, and temperature was 38.2 °C. A 22 G venous catheter (Delta ven 1, Delta Med, Viadana (MN), Italy) was placed in the left cephalic vein, and general anesthesia was induced with propofol (Proposure, 10 mg mL^−1^, Merial Italia S.p.A., Assago (MI), Italy) at 2 mg kg^−1^. Orotracheal intubation was achieved with a 3.5 mm diameter tube, and general anesthesia was maintained with isoflurane (Isoflo Zoetis S.r.l., Roma, Italy), titrated to effect, and delivered in oxygen at 100%. Every five minutes the depth of anesthesia was assessed, and the isoflurane vaporized settings were readjusted to achieve an adequate depth of anesthesia, characterized by absent palpebral reflex, ventromedial eye position, and relaxed jaw tone. During anesthesia, the patient presented discordant breathing, although the pulse oximeter detected oxygen saturation between 98 and 100%. Despite the lack of literature on pulse oximeter validation in this species, these values can be considered physiologic, as *Hapalemur alaotrensis* belongs to the mammalian group [13].

## 3. Results

### 3.1. Hematobiochemical Analisys

Blood samples obtained from the sedated patient, from the lateral saphenous vein, were largely unremarkable except for a mild lymphopenia (0.97 × 103 cells/µL; range 1.31–9.19 × 103 cells/µL) and increased hematocrit (48%; range 30–45%). It was not possible to compare albumin, cholesterol, and triglycerides measurement results with any published data (albumin, 3.2 g/dL; cholesterol, 207 mg/dL; triglycerides, 60 mg/dL). Complete blood count (CBC) was performed with the automated hematology analyzer Sysmex XT-2000iV (Sysmex, Kobe, Japan) using the “monkey” profile supplied by the instrument’s multispecies software [14]. Biochemical analyses were carried out with the automated spectrophotometer BT3500 (Biotecnica Instruments, Rome, Italy).

### 3.2. Radiographic Study

Chest and abdomen radiograms in right lateral and dorsal recumbency revealed a severely enlarged cardiac silhouette, which occupied the thoracic cavity from the third to the tenth pair of ribs, touching the thoracic wall bilaterally. It caused severe dorsal deviation of the trachea, lung, and diaphragmatic compression (Figure 1).

### 3.3. Echocardiographic Evaluation

Emergency echocardiography and echo-Doppler examinations, but not a thoracic focused assessment with sonography for trauma (T-FAST) evaluation, were performed, according to the American Society of Echocardiography guidelines, with an Esaote device (My Lab 50 Gold Cardiovascular) and a multifrequency phased-array transducer (7.5–10 MHz) [15]. No echocardiographic reference parameters for this species are reported in the literature, so they have been considered appropriate against those reported in human pediatrics patients of the same weight [16]. Abundant pericardial effusion without diastolic collapse of the right ventricle and no alterations in cardiac morphology were observed, justifying a chronic accumulation of pericardial fluid (Figure 2). A reduction of the fractional shortening, related to the use of alpha two agonist in premedication, was observed [10]. From the cranial to the cardiac base, adhering to the left costal wall at the level of the cranial mediastinum, there was a hyperechogenic neoformation with an inhomogeneous echo structure and irregular margins. Because of the difficulty in reaching the position, and the serious state of dyspnea of the subject, echo-guided, fine-needle cytology was not performed. Pericardiocentesis was performed, at the level of the right hemithorax between the fifth and the eighth intercostal spaces, under echocardiographic control, with drainage of 55 cc of milky whitish, slightly red-tinged, and turbid fluid. The fluid was placed in EDTA-containing tubes. Following pericardiocentesis, the subject developed a slight thoracic effusion, which was no longer present at the ultrasound check after a few weeks.

### 3.4. Cytological Examination

A cytological examination was performed on smears obtained both with line preparation and cytocentrifuge methods, and stained with May–Grunwald Giemsa. The cellular content of the fluid was of 3.43 × 103 nucleated cells/µL. Within a moderate hematic background, a mixed cell population consisting predominantly of small lymphocytes (85%), macrophages often containing hemosiderin, erythrocytes, and lipid droplets (10%) was identified; neutrophils, eosinophils, and plasma cells were also present in the fluid to a lesser extent (5%). No evidence of cytologic malignancies was identified (Figure 3). The biochemical characteristics of the transudate were as follows: total protein concentration (TP), 5.2 g/dL; total cholesterol, 153 mg/dL; triglyceride, 693 mg/dL. Both the cytological and the biochemical features classified the effusion as consistent with chylous [17,18]. The chemical-physical characteristics of the fluid recollected after two weeks, through a second pericardiocentesis, were superimposable to the previous one (TP, 5.6 g/dL; total cholesterol, 106 mg/dL; triglyceride, 914 mg/dL). On cytology, a mixed cell population of macrophages containing lipid droplets and, more rarely, erythrocytes and hemosiderin (about 60%), and small lymphocytes (about 40%) was observed and associated with a chronic chylous effusion.

### 3.5. CT Scan and LymphoCT

After the patient had been stabilized with pericardiocentesis, a CT scan was performed with a 16-slice CT scanner (GE Brightspeed^®^, General Electric Healthcare, Milano, Italy); whole body images were acquired before and after intravenous administration of non-ionic iodinated contrast media (iodixanol 320 mgI mL^−1^, at a dose of 2 mL kg^−1^). The CT scan revealed severe, homogeneous, non-enhancing, pericardial effusion (12 HU) and mild right pleural effusion that caused severe ventral lung compression. A cranial mediastinal mass was identified close to the thoracic wall, left-sided, localized at the level of the second intercostal space, ellipsoid-shaped (1 × 1.4 × 1.4 cm), with well-defined margins, soft tissue attenuation, and heterogeneously enhanced (Figure 4 and Figure 5).

LymphoCT by subcutaneous perianal administration of contrast medium, as described in the literature [19], was performed for the first time in this species to highlight the thoracic duct and identify any leakage. Mesenteric and popliteal lymph nodes were too small (respectively, 3 and 2 mm in diameter) to allow intranodal injection of contrast medium [20]. Contrast medium was injected subcutaneously into four different sites around the anus, then administration sites were gently massaged for five minutes. CT single slice scans at mid abdomen level were performed 5, 10, 15, and 30 min after massage, and one last scan was performed after echo-guided pericardiocentesis (about 60 min after massage). No migration of contrast medium from the injection sites was observed; therefore, lymphatic vessels, cisterna chyli, and the thoracic duct could not be identified. Thoracic computed tomography (CT) and lymphoCT were unable to show any causes of the chylous pericardial effusion; therefore, a thoracoscopic pericardial window had been scheduled.

The day after pericardiocentesis and the CT scan, oral prednisolone therapy was started at 0.5 mg/kg twice daily until clinical control (Prednicortone^®^ 5 mg, Dechra, Torino, Italy). Given the herbivorous nature of the patient, no dietary therapy with a restrictive diet was established. For the first fifteen days the subject did not manifest dyspnea, but then the clinical signs reappeared, and the animal was returned to the Veterinary Teaching Hospital. Despite the steroid therapy, during the second echocardiographic control the presence of abundant pericardial fluid was still evident. About 70 mL of pericardial fluid was drained through pericardiocentesis. During the second echocardiographic control, it was possible to perform an ultrasound-guided cytology of the mediastinal mass, but the cytological examination was inconclusive. The subject died spontaneously one week after the second pericardiocentesis.

### 3.6. Postmortem Examination

Postmortem examination highlighted conjunctival, oral, and vulvar cyanosis. At skinning, there was severe and diffuse skeletal muscles congestion. The severely dilated pericardial sac almost completely occupied the thoracic cavity, causing severe and bilateral pulmonary compression (Figure 6). About 100 mL of yellowish-white, slightly turbid liquid was drained from the pericardial sac, where a fibrin clot was also identified. In the cranial mediastinum, surrounded by a moderate amount of adipose tissue, a severely enlarged lymph node was detected (Figure 7). In the abdominal cavity, along with moderate spleno- and hepatomegaly, the caudal vena cava was severely congested, and numerous hemorrhagic suffusions and petechiae were observed in the small intestine. Further pathological findings were congestion of jugular veins and severe, diffuse meningeal hyperemia.

Histology showed severe pulmonary hyperemia and atelectasis, associated with pulmonary edema, alveolar hemorrhages, rare intra-alveolar hemosiderophages, and mild multifocal compensatory emphysema (Figure 8). The thoracic duct was not identified during necropsy. Microscopic examination of the heart failed to detect significant pathological changes. The mediastinal lymph node was microscopically characterized by edema and severe hyperemia associated with a locally extensive hemorrhage (Figure 9). The hemorrhagic nature of intestinal lesions was confirmed histologically. Microscopic examination identified severe and diffuse hyperemia of hepatic centrolobular veins and allowed diagnosis as adenoma of the rete ovarii, a 4 mm nodule detected in the right ovary. Anatomopathological examination identified a cardiogenic shock, secondary to pressure changes developed in the thoracic cavity, as the most likely cause of death of the subject. This hypothesis is supported by the severe multi-organ congestion, caudal vena cava and jugular veins distension, and cyanosis of external mucous membranes.

Eventually, a PCR test for encephalomyocarditis virus was also performed at the virology laboratory of Istituto Zooprofilattico Sperimentale della Lombardia e dell’Emilia Romagna (Brescia, Italy) on a pericardial effusion sample. This research has not demonstrated the presence of this causative agent (Cardiovirus).

## 4. Discussion

To the authors’ knowledge, this is the first report that describes the anesthetic protocol and the diagnostic trial for the diagnosis of chylopericardium in a *Hapalemur alaotrensis*. Furthermore, this is the first description of clinical use of CT lymphography of the thoracic duct by perianal subcutaneous injection of iodinated contrast medium in a lemur. 

Unfortunately, no migration of subcutaneously injected contrast medium was observed, and it was not possible to identify the thoracic duct nor any connection between it and the pericardial sac. This is likely attributable to limitations of the lymphoCT procedure rather than the absence of a connection between these two structures. In fact, in our experience we observed migration of contrast medium in almost half of the patients (dogs and cats) that underwent CT lymphography of the thoracic duct by perianal subcutaneous injection of iodinated contrast medium. 

## 5. Conclusions

In conclusion, this case report confirms that it is possible that chylopericardium is a self-perpetuating condition for which pericardial effusion may lead to a chronic increase in right-sided venous pressure that prevents the normal forward flow of chylous into the cranial vena cava. The final diagnosis was performed by pericardial fluid analysis, and unfortunately, it was not possible to detect a rupture of the thoracic duct through the additional diagnostic investigations described. 

Furthermore, the conservative therapy and the effective drainage resulted in a therapeutic failure, superimposable to a failure rate of 57–60% reported for this kind of approach.

We believe that the described procedures, the anesthesiologic protocol, the cytological examination of the effusion, the CT examination, and the anatomopathological description may be useful for the colleagues who are faced with the management of a case of chylopericardium in unconventional species.

## Figures and Tables

**Figure 1 animals-11-00536-f001:**
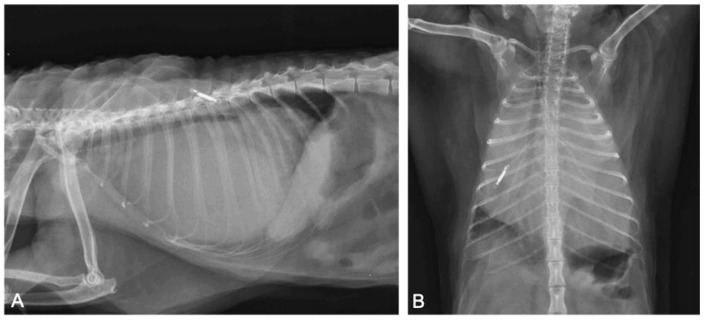
Thoracic radiographs ((**A**): right-lateral view; (**B**): ventro-dorsal view). The cardiac silhouette appears severely enlarged, causing dorsal displacement of the trachea and lung and diaphragmatic compression. The microchip transponder is on the left side.

**Figure 2 animals-11-00536-f002:**
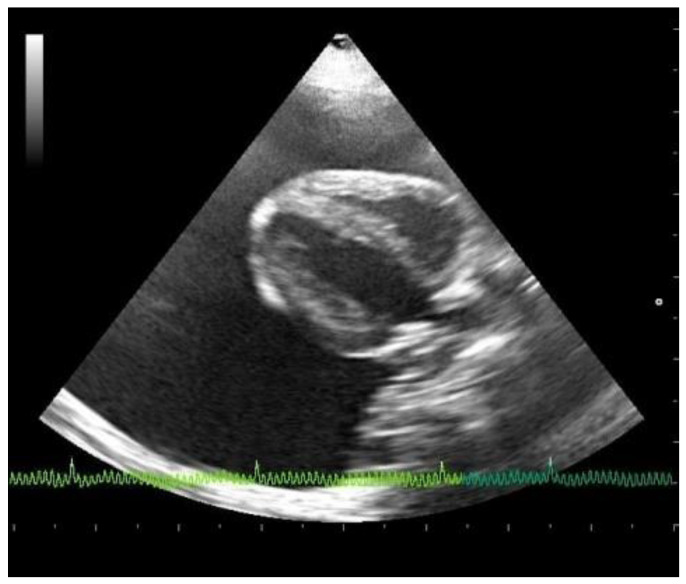
Echocardiographic right parasternal B-mode evaluation (four chambers view) showing the presence of severe pericardial effusion.

**Figure 3 animals-11-00536-f003:**
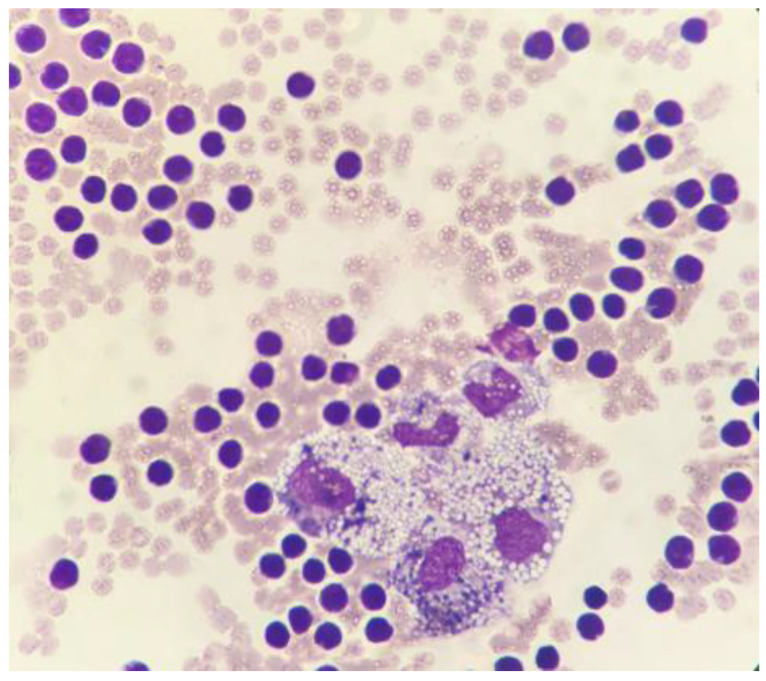
Cytocentrifuged specimen of the pericardial effusion. A moderate hematic background is present along with a mixed cell population consisting mainly of small lymphocytes and macrophages containing both lipid droplets and hemosiderin. May–Grunwald Giemsa, 100×.

**Figure 4 animals-11-00536-f004:**
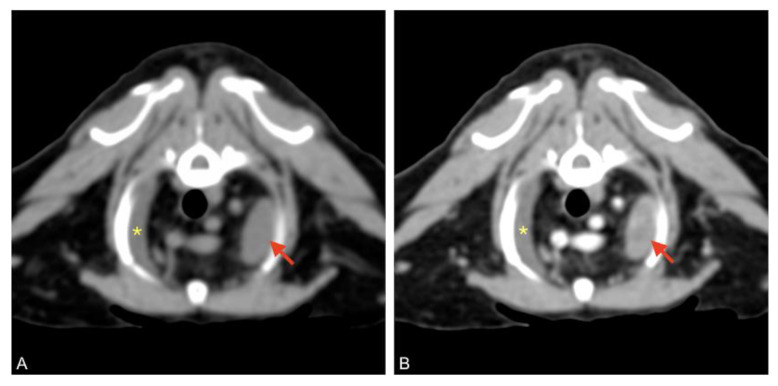
Pre (**A**) and post-contrast (**B**) images of the left-sided cranial mediastinal mass (red arrows), adjacent to the thoracic wall, ellipsoid-shaped, and heterogeneously enhanced. Mild pleural effusion is visible on the right (asterisk).

**Figure 5 animals-11-00536-f005:**
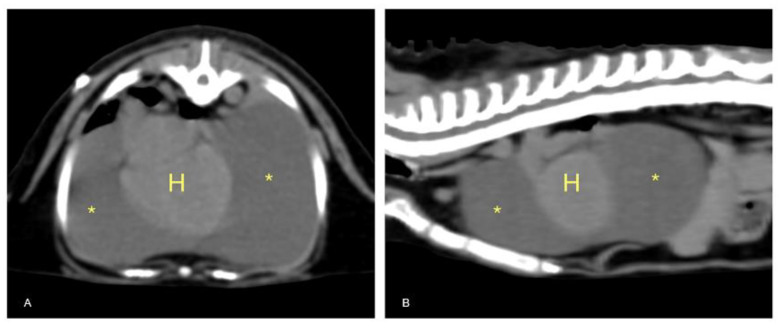
Transverse (**A**) and sagittal (**B**) TC sections of the thorax. Severe pericardial effusion (asterisks) with mild right dorsal-lateral heart (H) displacement.

**Figure 6 animals-11-00536-f006:**
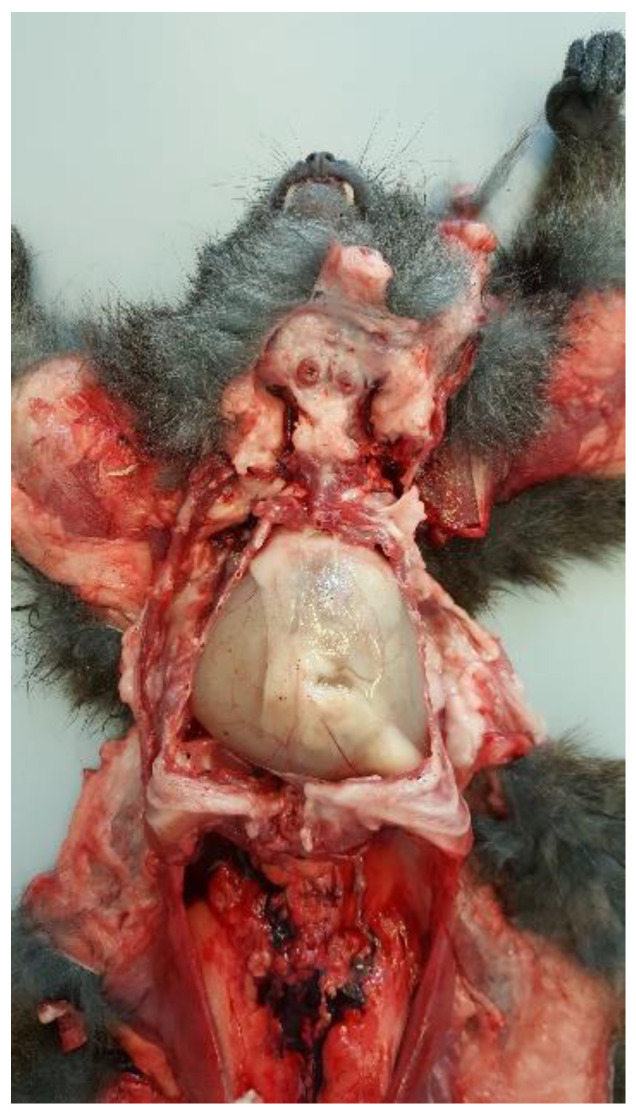
Anatomopathological examination of the thoracic cavity. There is a severe dilatation of the pericardial sac with severe bilateral lung compression.

**Figure 7 animals-11-00536-f007:**
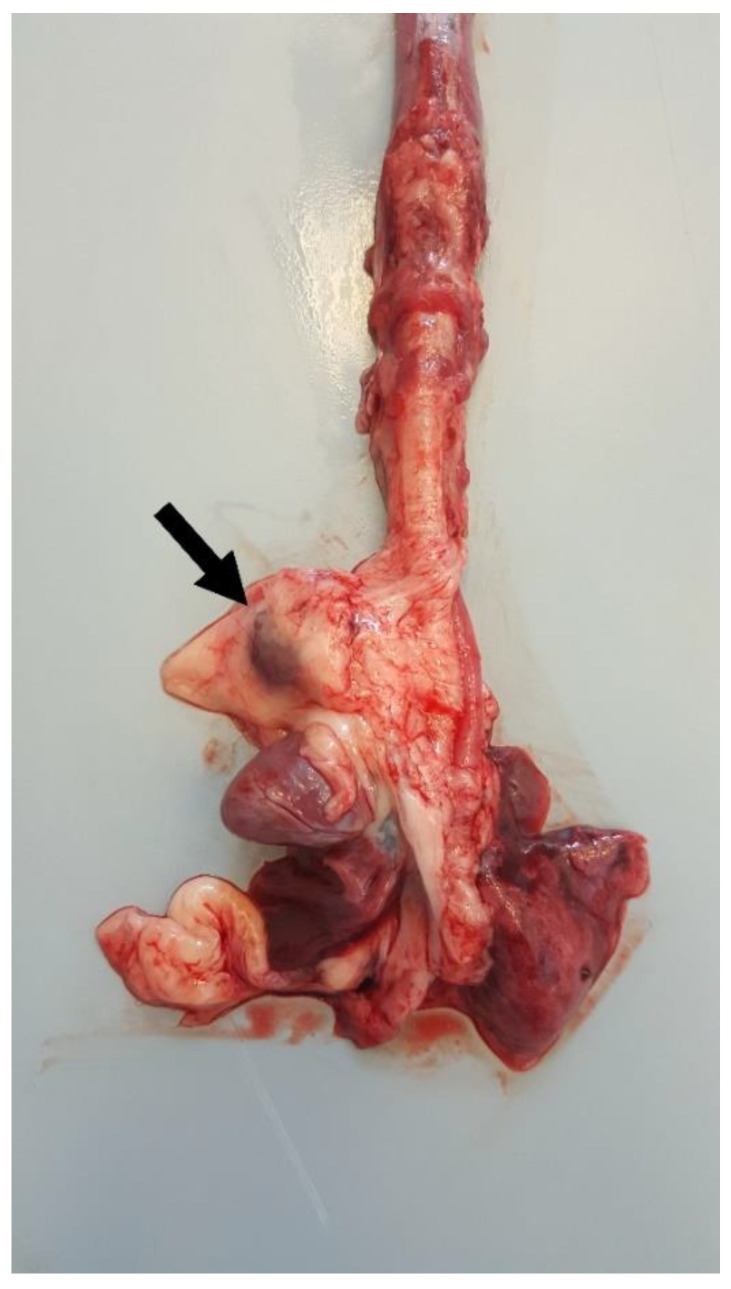
Thoracic organs. In the cranial mediastinum, surrounded by a moderate amount of adipose tissue, a severely enlarged lymph node (arrow) was detected.

**Figure 8 animals-11-00536-f008:**
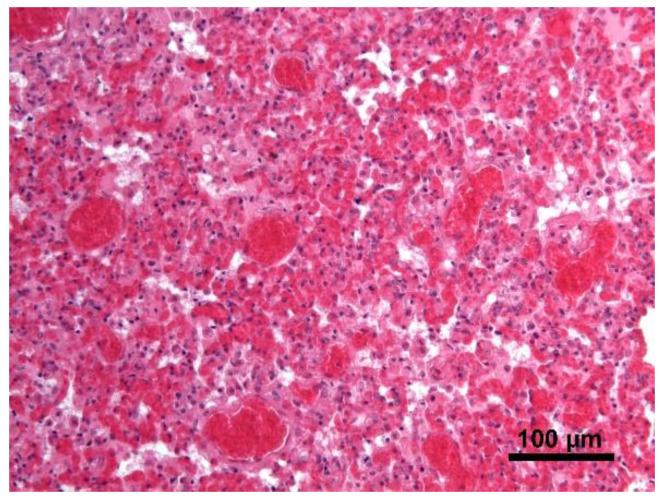
Lung. Severe pulmonary hyperemia and atelectasis; Hematoxylin and eosin stain, 20× magnification.

**Figure 9 animals-11-00536-f009:**
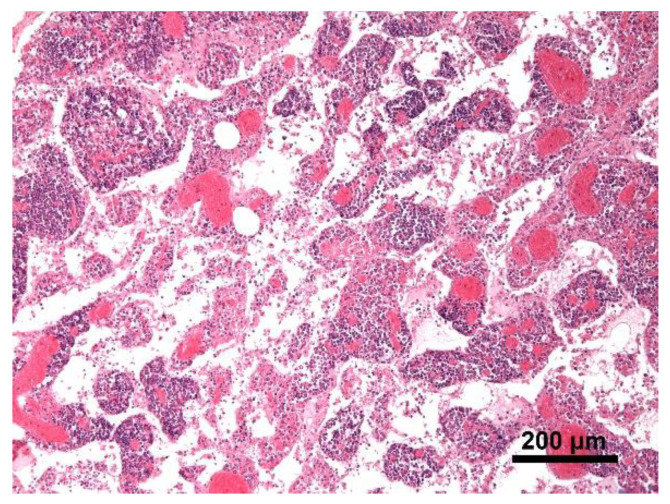
Mediastinal lymph node. The lymph node was microscopically characterized by edema and severe hyperemia; Hematoxylin and eosin stain, 10× magnification.

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
