# Peer review of "Chylopericardium Effusion in a Lac Alaotra Bamboo Lemur (Hapalemur alaotrensis)"

_animals, 2021, doi:10.3390/ani11020536_

Round 1

Reviewer 1 Report

-I suggest to modify in “chylopericardium effusion in a Lac Alaotra Bamboo Lemur (Hapalemur Alaotrensis)”

-I can’t see the difference between the pre e post contrast images in fig 3. I suggest to improve the quality of the pictures and add the description of the red arrow in the legend.

-with the TC the authors declared to detect two masses: why they didn’t described it during the post mortem examination? I think it was also interesting to improve the quality of pictures of necropsy with other findings, such as pictures of these masses or other pictures of the thoracic cavity with the other findings (enlarged lymph node/lung lesions...) and/ or adding a picture of the histology for completeness.

Author Response

The authors thank the Editor and the Reviewers for their second thoroughly and rapid review of our manuscript.

We have carefully considered all reviewers’ comments and have tried to address them, clarifying some misunderstanding. We are very grateful of this constructive exchange of opinions and comments.

Best regards

Reviewer 1

Comments and Suggestions for Authors

-I suggest to modify in “chylopericardium effusion in a Lac Alaotra Bamboo Lemur (Hapalemur Alaotrensis)”

Thank you for your suggestion, we have modified the title. In this way the topic is clearer to the reader. Thank you.

-I can’t see the difference between the pre e post contrast images in fig 3. I suggest to improve the quality of the pictures and add the description of the red arrow in the legend.

Thank you, we have changed this figure with one with more detail and we have modified the caption.

-with the TC the authors declared to detect two masses: why they didn’t described it during the post mortem examination? I think it was also interesting to improve the quality of pictures of necropsy with other findings, such as pictures of these masses or other pictures of the thoracic cavity with the other findings (enlarged lymph node/lung lesions...) and/ or adding a picture of the histology for completeness.

Thank you very much for this comment. We have decided to add some images of histological examination of lungs and lymph node in order to be more complete in our description. Furthermore, due to the impossibility to observe and describe the second mass during the post-mortem examination, we decided to delete this information in the CT and lymphoCT description. Thank you.

Reviewer 2 Report

This manuscript describes interesting report for chylopericardium in bamboo lemur (Hapalemur alaotrensis).

It is also interesting to note that this bamboo lemur had normal lungs and pleural space.

Authors have provided enough background information on pathophysiology of chylopericardium. It would be nice if authors provide  some  information about chylothorax in cats.

Authors provided good clinical data. It would be interesting to see vitals (temp, HR, RR) at the presentation.

It is also important to rule out any infectious cause of presentation. It would nice to add radiographs of the case to the manuscript. Authors have not mentioned if they have done T-FAST.

Overall, this is good clinical report.

Author Response

The authors thank the Editor and the Reviewers for their second thoroughly and rapid review of our manuscript.

We have carefully considered all reviewers’ comments and have tried to address them, clarifying some misunderstanding. We are very grateful of this constructive exchange of opinions and comments.

Best regards

Reviewer 2

Comments and Suggestions for Authors

This manuscript describes interesting report for chylopericardium in bamboo lemur (Hapalemur alaotrensis).

It is also interesting to note that this bamboo lemur had normal lungs and pleural space.

Authors have provided enough background information on pathophysiology of chylopericardium. It would be nice if authors provide some information about chylothorax in cats.

Thank you for this suggestion. We added a citation regarding the treatment and the outcome of Chylopericardium in dogs and in cats.

Authors provided good clinical data. It would be interesting to see vitals (temp, HR, RR) at the presentation.

Thank you, we explain the lack with this sentence: “To perform clinical procedures, general anesthesia was necessary. For this reason, it was not possible to record the basal vital parameters of the subject.” Furthermore, we have described the vitals after the premedication. Thank you for this comment.

It is also important to rule out any infectious cause of presentation.

We agree with the reviewer regarding this matter. Hematology and biochemical results were not highly suggestive of an ongoing infectious disease due to the absence of specific inflammatory response (leukocyte count increase vs degenerative left shift, lymphocytosis, thrombocytosis, total protein increase with A/G ratio decrease as well as any biochemical alterations indicative of specific organ involvement). Moreover, the presence of a virus that could have been responsible for the patient’s clinical presentation (encephalomyocarditis virus) was excluded via molecular testing. Performing additional PCR tests for several other microorganisms would have been cost-prohibitive and difficult to interpret, since the presence of an antigen does not always correlate with diagnosis. If the reviewer thinks it is necessary, authors will be happy to add a sentence regarding the fact that unfortunately other infectious causes were not excluded.

It would nice to add radiographs of the case to the manuscript.

Thank you, we have added the figure with the radiographic examination. In this way the presentation is more precise. Thank you.

Authors have not mentioned if they have done T-FAST.

Thank you, unfortunately we have not performed it. We added this sentence: “Emergency echocardiography and echo-Doppler examination, but not a T-FAST evaluation, were performed”.

Overall, this is good clinical report.

We are very pleasure for this comment. Thank you very much.
